

# Association of polymorphic markers of genes *FTO, KCNJ11, CDKAL1, SLC30A8,* and *CDKN2B* with type 2 diabetes mellitus in the Russian population

Aleksey G. Nikitin[1,*], Viktor Y. Potapov[2,*], Olga I. Brovkina[1,*], Ekaterina O. Koksharova[3,*], Dmitry S. Khodyrev[1,*], Yury I. Philippov[3,*], Marina S. Michurova[3,*], Minara S. Shamkhalova[3,*], Olga K. Vikulova[3,4,*], Svetlana A. Smetanina[5,*], Lyudmila A. Suplotova[5,*], Irina V. Kononenko[3,4,*], Viktor Y. Kalashnikov[3,*], Olga M. Smirnova[3,4,*], Alexander Y. Mayorov[3,4,*], Valery V. Nosikov[6,*], Alexander V. Averyanov[1,*] and Marina V. Shestakova[3,4,*]

[1] Federal Research Clinical Center for Specialized Types of Health Care and Medical Technologies of Federal Medical and Biology Agency, Moscow, Russian Federation
[2] Clinic of New Medical Technologies "Archimedes", Moscow, Russian Federation
[3] Endocrinology Research Centre, Moscow, Russian Federation
[4] I.M. Sechenov First Moscow State Medical University, Moscow, Russian Federation
[5] Tyumen State Medical University, Tyumen, Russian Federation
[6] State Research Institute of Genetics and Selection of Industrial Microorganisms, Moscow, Russian Federation
[*] These authors contributed equally to this work.

Corresponding author
Yury I. Philippov,
yuriyivanovich@gmail.com

## ABSTRACT

**Background**. The association of type 2 diabetes mellitus (T2DM) with the *KCNJ11, CDKAL1, SLC30A8, CDKN2B,* and *FTO* genes in the Russian population has not been well studied. In this study, we analysed the population frequencies of polymorphic markers of these genes.

**Methods**. The study included 862 patients with T2DM and 443 control subjects of Russian origin. All subjects were genotyped for 10 single nucleotide polymorphisms (SNPs) of the genes using real-time PCR (TaqMan assays). HOMA-IR and HOMA-$\beta$ were used to measure insulin resistance and $\beta$-cell secretory function, respectively.

**Results**. The analysis of the frequency distribution of polymorphic markers for genes *KCNJ11, CDKAL1, SLC30A8* and *CDKN2B* showed statistically significant associations with T2DM in the Russian population. The association between the *FTO* gene and T2DM was not statistically significant. The polymorphic markers *rs5219* of the *KCNJ11* gene, *rs13266634* of the *SLC30A8* gene, *rs10811661* of the *CDKN2B* gene and *rs9465871, rs7756992* and *rs10946398* of the *CDKAL1* gene showed a significant association with impaired glucose metabolism or impaired $\beta$-cell function.

**Conclusion**. In the Russian population, genes, which affect insulin synthesis and secretion in the $\beta$-cells of the pancreas, play a central role in the development of T2DM.

## INTRODUCTION

Diabetes mellitus is a group of metabolic diseases characterised by chronic hyperglycemia resulting from impaired insulin secretion, resistance to insulin, or both. Chronic hyperglycemia, due to underlying diabetes, is accompanied by impairment or dysfunction of various organs, particularly the eyes, kidneys, nerves, heart and blood vessels.

Type 2 diabetes mellitus (T2DM) is 10 times more common than type 1 diabetes mellitus. An epidemic of T2DM is occurring in every country of the world, particularly in industrialised countries. The prevalence of the disease varies in different regions, depending on the ethnicity of the population. According to the World Health Organization, T2DM is present in 3%–6% of the population in European countries, 5% of the population in the United States, 10% of African Americans, 24% of Americans of Mexican origin and 35% of the population of Micronesia and Polynesia (*World Health Organization, 2016a*).

The causes of T2DM pathogenesis include: insulin resistance, impairment of insulin secretion, an increase in the amount of glucose produced by the liver, genetic susceptibility, sedentary lifestyle and excessive caloric intake that leads to obesity. Heredity undoubtedly plays a crucial role in the development of T2DM, with lifestyle exacerbating genetically determined insulin resistance (IR) (*World Health Organization, 2016b*).

T2DM has a polygenetic nature, i.e., the clinical phenotype is a result of the effects of several genetic loci (*Wang et al., 2016*). Currently, approximately 70 genes have been identified whose variants predispose one to the development of T2DM (*Hollensted et al., 2016*; *Hara et al., 2014*). However, susceptibility varies across populations due to ethnic differences in the polymorphisms, variations in the structure of the haplotypes/linkage disequilibrium blocks and the influence of non-genetic factors. These genes can be divided into two types based upon their contribution to development of diabetes: genes associated with the impairment of development, growth, proliferation and functioning of the β-cells of the pancreas, and genes that affect the development of insulin resistance in peripheral tissues, such as muscles and liver.

Mutations in the *KCNJ11* gene, which is located at 2q36, may be associated with the development of T2DM, due to impaired regulation of insulin from the β-cells of the pancreas. The Kir6.2 protein encoded by this gene is one of two subunits that form a channel for potassium ions (*Aguilar-Bryan & Bryan, 1999*). ATP-dependent potassium channels take part in the regulation of insulin secretion through changes in the cell membrane potential of the β-cells. Mutations in the *KCNJ11* gene lead to changes in the structure of the Kir6.2 channel and may lead to neonatal diabetes and congenital hyperinsulinemia (*Albaqumi et al., 2014*; *Gohar et al., 2016*). The rs5219 polymorphism in exon 1 of the *KCNJ11* gene has been associated with the development of T2DM (*Sakura et al., 1996*). This polymorphism has been associated with a reduction of insulin secretion in individuals with normal glucose levels (*Nichols, Koster & Remedi, 2007*).

Cyclin-dependent kinase inhibitors constitute a family of proteins that regulate cell cycle, cell proliferation and differentiation. Impaired functioning of these proteins is associated with the development of cancer, ischaemic heart disease and diabetes mellitus (*Fajas, Blanchet & Annicotte, 2010*). The *CDKN2A/2B* genes, which are located at 9p21, are expressed in all cells, including adipocytes and pancreatic β-cells. Studies in muscle cells have shown that the protein encoded by the *CDKN2B* gene affects insulin secretion through regulation of the expression of the *E2F1* gene (*Kim & Rane, 2011*). The *CDKN2A* gene is likely to be involved in the development of T2DM through an age-dependent reduction in the number and regenerative potential of β-cells, leading to the overall deterioration of the endocrine function of the pancreas (*Tschen et al., 2009*).

The *CDKAL1* gene, located at 6p22.3, is homologous to the CDK5RAP1 inhibitor of the CDK5 kinase (*Hurst et al., 2008*). It has been shown that *CDKAL1* also acts as an inhibitor in pancreatic β-cells; CDK5 kinase activity plays a significant role in the efficiency of insulin granule secretion into the bloodstream (*Wei et al., 2005*; *Ubeda, Rukstalis & Habener, 2006*).

One of the major causes of T2DM development is a reduction in insulin secretion. This process requires the optimal concentration of zinc ions in the β-cells of the pancreas, which are regulated by type 8 zinc carrier proteins (ZnT8) (*Dunn, 2005*). ZnT8is encoded by the *SLC30A8* gene located near 8q24.11. The expression of this gene is most intense in pancreatic β-cells (*Smidt et al., 2016*). The participation of the *SLC30A8* gene in the development of T2DM has been substantiated in several large-scale studies (*Saxena et al., 2007*; *Horikawa et al., 2008*; *Ng et al., 2008*).

The *FTO* gene is located at 16q12.2. Its function in the development of obesity remains to be determined. The *FTO* gene is expressed in various tissues, particularly the hypothalamus, liver, muscle tissue, adipocytes and the β-cells of the pancreas (*Stratigopoulos et al., 2008*). Its expression in the subcutaneous fat is higher than in other tissues, although its expression in other tissues that affects the body mass index (BMI) (*Kloting et al., 2008*).

This study examined the association of the polymorphic markers of the genes *KCNJ11*, *SLC30A8*, *CDKAL1*, *CDKN2B* and *FTO* with type 2 diabetes mellitus in Russia. These polymorphisms have produced controversial results in studies on several European populations. The data in the current literature for these genes is very limited.

## MATERIALS AND METHODS

The study compared 862 patients diagnosed with T2DM (DM2+) to a control group (DM2−) consisting of 443 randomly selected patients showing no signs of T2DM based on clinical and biochemical examinations. Subjects of the DM2+ group were patients at the Endocrinology Research Center (Moscow, Russia) and Tyumen State Medical University (Tyumen, Russia) and were of European ancestry, based upon the results of a questionnaire. The groups were similar in terms of age and sex (Table 1).

Blood glucose and insulin concentrations were measured at baseline and two h after an oral glucose tolerance test. The homeostasis model assessment of insulin resistance (HOMA-IR) and the homeostasis model assessment of β-cell function (HOMA- β) indices

**Table 1** Characteristics of the examined groups.

| Characteristics | DM2+ ($n = 862$) | DM2− ($n = 443$) |
|---|---|---|
| Age (years) | $60.0 \pm 10.2$ | $54.4 \pm 11.0$ |
| BMI[a] | $30.5 \pm 5.0$ | $28.7 \pm 4.8$ |
| Basal glucose level (mol/l) | $9.4 \pm 1.3$ | $5.1 \pm 0.7$ |
| Glucose level 2 h after PGTT[b] (mol/l) | $12.1 \pm 1.4$ | $6.9 \pm 0.8$ |
| Basal insulin level (mU/l) | $14.9 \pm 5.4$ | $10.4 \pm 4.3$ |
| Insulin level 2 h after PGTT[b] (mU/l) | $93.6 \pm 28.4$ | $41.9 \pm 10.3$ |
| Glycated hemoglobin HBA1C (%) | $7.4 \pm 1.9\%$ | – |
| HOMA-b | $47.8 \pm 16.1$ | $94.3 \pm 30.6$ |
| HOMA-IR | $6.7 \pm 1.3$ | $2.8 \pm 1.5$ |

**Notes.**

[a] BMI—body mass index.

[b] PGTT—peroral glucose tolerance test.

were calculated for the purpose of evaluating the insulin resistance in tissues and β-cell function, respectively (*Matthews et al., 1985*). Genomic DNA was phenol-chloroform extracted from whole blood samples after incubation with proteinase K in the presence of 0.1% sodium dodecyl sulfate using conventional methods (*Johns & Paulus-Thomas, 1989*).

Real-time PCR was used to amplify regions of interest within the target genes. PCR was conducted using 50–100 ng of genomic DNA in 20 μL of a reaction mixture containing 70 mM Tris-HCI, pH 8.8, 16.6 mM ammonium sulfate, 0.01% Tween-20, 2 mM magnesium chloride, 200 nmol of each dNTP, 500 nmol primers (Evrogen, Moscow, Russia), 350 nmol of fluorescent probes (DNK-Sintez, Moscow, Russia) and 1.5 U Taq DNA-polymerase (Evrogen, Moscow, Russia). Amplification was carried out using an StepOnePlus thermal cycler (Applied Biosystems, Forster City, CA, USA) using the following conditions: initial denaturation at 95 °C for two min; 40 cycles of denaturation (94 °C) for 10 s, annealing (54 °C–66 °C) for 60 s, extension (72 °C) for 10 s. The fluorescent dyes used in the probes were carboxyfluorescein and hexachlorofluorescein, and the fluorescence extinguisher was BHQ-1. The sequences of primers, fluorescent probes and the method for determining the genotypes of the examined loci are presented in supplementary Table S1. Designations of polymorphic markers comply with the standards of the dbSNP database (http://www.ncbi.nlm.nih.gov/snp/).

The genotype analysis of polymorphic markers of several genes was performed through endpoint fluorescence detection using the built-in tools of the SDS 2.3 software, with a sample considered positive if its quality value was 95%. Samples that failed to meet this quality value were re-analysed (100% of samples were subjected to genotype analysis). Contingency tables and chi-square tests were used for statistical analyses of the allelic distributions of the SNPs in the DM2+ and DM2− groups. Calculations were performed using the calculator for statistical computation in case-control studies (*Gene Expert, 2013*) and SPSS, ver. 17. For all analyses, $P < 0.05$ was considered to be statistically significant. Analysis of variance was used to test for associations between gene polymorphisms and metabolic characteristics (glucose and insulin levels, HOMA-IR and HOMA-β indices).

Genes that exhibited no reliable or reproducible data for the Russian population were selected to determine any association. Due to the conflicting results obtained by other researchers, the examination of the entire linkage disequilibrium block in the promoter region of the *FTO* gene was investigated. HaploView 3.2 was used for the analysis of linkage disequilibrium blocks and selection of polymorphic markers for the *FTO* gene (*Barrett et al., 2005*).

The local Committee for Ethics of Endocrinology Research Centre (Moscow, Russian Federation) granted ethical approval for the study (Ethical Application Ref: protocol No.14AB on 27-nov-2014).

## RESULTS

The prevalence of alleles of polymorphic markers of *FTO*, *KCNJ11*, *CDKAL1*, *SLC30A8* and *CDKN2B* in the sample population was not significantly different from the prevalence in a typical European population (data for the European population was obtained from the HapMap (CEU) project: http://hapmap.org). The distribution of alleles in DM2+ and DM2− groups was consistent with the distribution predicted by the Hardy-Weinberg equilibrium, which permitted the use of a multiplicative inheritance model for the analysis of associations between polymorphic markers and metabolic phenotypes (*Lewis, 2002*).

Table 2 summarises the results of the analysis of associations of the examined markers with T2DM. The following polymorphic markers showed statistically significant association with T2DM: *rs5219* of the *KCNJ11* gene, *rs13266634* of the *SLC30A8* gene, *rs10811661* of the *CDKN2B/2A* gene, *rs9465871*, *rs7756992* and *rs10946398* of the *CDKAL1* gene.

Table 3 summarises the results of the association analysis for the examined SNPs and metabolic indicators of glucose intolerance and β-cell dysfunction. All results with $P < 0.05$ for at least one indicator are shown. The following polymorphic markers showed a significant association with impaired glucose metabolism or impaired β-cell function: *rs5219* of the *KCNJ11* gene, *rs13266634* of the *SLC30A8* gene, *rs10811661* of the *CDKN2B* gene and *rs9465871*, *rs7756992* and *rs10946398* of the *CDKAL1* gene.

## DISCUSSION

The *KCNJ11* gene contains the SNP *rs5219* in exon 1 (substitution G →A), which leads to the substitution of Glu for Lys at position 23. Although several studies on the association of this polymorphism with T2DM in different populations have produced conflicting results (*Scott et al., 2007*), more recent studies have found an association between this polymorphic marker and the disease (*Salonen et al., 2007*). Increased numbers of patients in study populations have revealed an association between this polymorphic marker and the T2DM development (*Shaat et al., 2005*; *Florez et al., 2007*; *Sakamoto et al., 2007*; *Gonen et al., 2012*; *Iwata et al., 2012*; *Odgerel et al., 2012*; *Phani et al., 2014*). Despite the fact that this association was found by other investigators (*Florez et al., 2004*), the *K23* allele has been associated with the increased risk of T2DM development in many European (odds ratio (OR) = 1.23) and Asian populations (OR = 1.26) (*Nielsen et al., 2003*). An analysis of the distribution of frequencies, alleles and genotypes of the polymorphic marker *rs5219*

Nikitin et al. (2017), PeerJ, DOI 10.7717/peerj.3414

**Table 2** Comparative analysis of allele and genotype distribution of polymorphic markers of the genes *FTO, KCNJ11, CDKAL1, SLC30A8,* and *CDKN2B.*

| Gene | Polymorphic marker | Genotype | Distribution of genotypes | | Model | | | | | |
|---|---|---|---|---|---|---|---|---|---|---|
| | | | DM2+ | DM2− | Multiplicative | | Dominant | | Recessive | |
| | | | N = 862 | N = 443 | p | OR (95% CI) | p | OR (95% CI) | p | OR (95% CI) |
| FTO | rs8050136 | C/C | 272 (0, 32) | 143 (0, 32) | 0.1 | 0.97 (0.76–1.24) | 0.79 | 0.97 (C/C) (0.76–1.24) 1.04 (C/A+A/A vs. C/C) (0.81–1.32) | 0.02 | 1.76 (A/A) (1.04–2.98) 0.57 (C/C+C/A vs. A/A) (0.34–0.96) |
| | | C/A | 527 (0, 61) | 281 (0, 63) | | 0.91 (0.72–1.15) | | | | |
| | | A/A | 63 (0, 07) | 19 (0, 04) | | 1.76 (1.04–2.98) | | | | |
| | rs7202116 | A/A | 225 (0, 26) | 124 (0, 28) | 0.72 | 0.91 (0.70–1.18) | 0.47 | 0.91 (A/A) (0.70–1.18) 1.10 (A/G+G/G vs. A/A) (0.85–1.42) | 0.91 | 0.98 (G/G) (0.74–1.31) 1.02 (A/A+A/G vs. G/G) (0.76–1.36) |
| | | A/G | 468 (0, 54) | 231 (0, 52) | | 1.09 (0.87–1.37) | | | | |
| | | G/G | 169 (0, 2) | 88 (0, 2) | | 0.98 (0.74–1.31) | | | | |
| | rs9930506 | A/A | 208 (0, 24) | 115 (0, 26) | 0.67 | 0.91 (0.70–1.18) | 0.47 | 0.91 (A/A) (0.70–1.18) 1.10 (A/G+G/G vs. A/A) (0.85–1.43) | 0.47 | 1.11 (G/G) (0.68 –1.20) 0.90 (A/A+A/G vs. G/G) (0.84 –1.47) |
| | | A/G | 466 (0, 54) | 239 (0, 54) | | 1.00 (0.80–1.26) | | | | |
| | | G/G | 188 (0, 22) | 89 (0, 2) | | 1.11 (0.84–1.47) | | | | |
| KCNJ11 | rs5219 | Glu/Glu | 174 (0, 2) | 124 (0, 28) | 0.0007 | 0.65 (0.50–0.85) | 0.001 | 0.65 (Glu/Glu) 1.54 (0.50–0.85) (Glu/Lys + Lys/Lys vs. Glu/Glu) (1.18–2.01) | 0.004 | 1.55 (Lys/Lys) (0.48–0.87) 0.64 (Glu/Glu + Glu/Lys vs. Lys/Lys) (1.15–2.09) |
| | | Glu/Lys | 486 (0, 56) | 246 (0, 56) | | 1.04 (0.82–1.30) | | | | |
| | | Lys/Lys | 202 (0, 23) | 73 (0, 16) | | 1.55 (1.15–2.09) | | | | |
| SLC30A8 | rs13266634 | C/C | 449 (0, 52) | 268 (0, 6) | 0.004 | 0.71 (0.56–0.90) | 0.004 | 0.71 (C/C) (0.56–0.90) 1.41 (C/T + T/T vs. C/C) (1.12–1.78) | 0.01 | 1.86 (T/T) (1.13–3.06) 0.54 (C/C + C/T vs. T/T) (0.33–0.89) |
| | | C/T | 340 (0, 39) | 154 (0, 35) | | 1.22 (0.96–1.55) | | | | |
| | | T/T | 73 (0, 08) | 21 (0, 05) | | 1.86 (1.13–3.06) | | | | |
| CDKN2B | rs10811661 | T/T | 285 (0, 33) | 209 (0, 47) | 1.0E−7 | 0.55 (0.44–0.70) | 7.0E−7 | 0.55 (T/T) (0.44–0.70) 1.81 (T/C + C/C) (1.43–2.29) | 2.0E−5 | 2.10 (C/C) (1.49–2.97) 0.48 (T/T + T/C) (0.34–0.67) |
| | | C/T | 405 (0, 47) | 187 (0, 42) | | 1.21 (0.96–1.53) | | | | |
| | | C/C | 172 (0, 2) | 47 (0, 11) | | 2.10 (1.49 –2.97) | | | | |
| | rs7756992 | A/A | 390 (0, 45) | 235 (0, 53) | 0.0003 | 0.73 (0.58–0.92) | 0.008 | 0.73 (A/A) (0.58–0.92) 1.37 (A/G + G/G vs. A/A) (1.09–1.72) | 0.0001 | 2.06 (G/G) (1.42–3.00) 0.49 (A/A + A/G vs. G/G) (0.33–0.71) |
| | | A/G | 329 (0, 38) | 169 (0, 38) | | 1.00 (0.79–1.27) | | | | |
| | | G/G | 143 (0, 17) | 39 (0, 09) | | 2.06 (1.42–3.00) | | | | |
| | rs9465871 | C/C | 259 (0, 3) | 190 (0, 43) | 1.0E−5 | 0.57 (0.45–0.73) | 4.0E−6 | 0.57 (C/C) (0.45–0.73) 1.75 (C/T + T/T vs. C/C) (1.38–2.22) | 0.02 | 1.49 (T/T) (0.47–0.95) 0.67 (C/C + C/T) (1.05–2.12) |
| | | C/T | 468 (0, 54) | 204 (0, 46) | | 1.39 (1.11–1.75) | | | | |
| | | T/T | 135 (0, 16) | 49 (0, 11) | | 1.49 (1.05–2.12) | | | | |

Nikitin et al. (2017), *PeerJ*, DOI 10.7717/peerj.3414

**Table 2** (*continued*)

| Gene | Polymorphic marker | Genotype | Distribution of genotypes | | Model | | | | | |
|------|-------------------|----------|------------------|------------------|--------------------------|------------------|------------------|------------------|------------------|------------------|
| | | | DM2+ | DM2− | Multiplicative | | Dominant | | Recessive | |
| | | | N = 862 | N = 443 | *p* | *OR (95% CI)* | *p* | *OR (95% CI)* | *p* | *OR (95% CI)* |
| CDKAL1 | *rs7754840* | C/C | 440 (0, 51) | 205 (0, 46) | 0.26 | 1.21 (0.96–1.52) | 0.61 | 0.88 (G/G) (0.53–1.46) | 0.1 | 1.21 (C/C) (0.96–1.52) 0.83 |
| | | C/G | 379 (0, 44) | 213 (0, 48) | | 0.85 (0.67–1.07) | | 1.14 (C/C + | | (C/G + G/G) |
| | | G/G | 43 (0, 05) | 25 (0, 06) | | 0.88 (0.53–1.46) | | C/G vs. G/G) (0.69–1.89) | | (0.66–1.04) |
| | *rs10946398* | A/A | 500 (0, 58) | 297 (0, 67) | 0.004 | 0.68 (0.53–0.86) | 0.002 | 0.68 (A/A) (0.53–0.86) 1.47 | 0.04 | 1.67 (C/C) (1.02–2.73) 0.60 |
| | | A/C | 293 (0, 34) | 124 (0, 28) | | 1.32 (1.03–1.70) | | (A/C + C/C vs. | | (A/A + A/C vs. |
| | | C/C | 69 (0, 08) | 22 (0, 05) | | 1.67 (1.02–2.73) | | A/A) (1.16–1.87) | | C/C) (0.37–0.98) |

**Table 3  Analysis of associations of polymorphic markers of the genes *FTO, KCNJ11, CDKAL1, SLC30A8, and CDKN2B* with the metabolic indicators of glucose tolerance and β-cell function.**

| Gene | Polymorphic marker | Genotype | Insulin level 2 h after PGGT** (mU/l) | | | HOMA-β | | |
|---|---|---|---|---|---|---|---|---|
| | | | DM2+ N = 862 | DM2− N = 443 | p (DM+/DM−) | DM2+ N = 862 | DM2− N = 443 | p (DM+/DM−) |
| FTO | rs8050136 | C/C | 80.9 ± 24.9 | 51.2 ± 24.9 | −/− | 59.2 ± 24.3 | 99.2 ± 36.1 | −/− |
| | | C/A | 78.7 ± 32.2 | 49.8 ± 25.2 | | 56.3 ± 22.4 | 99.3 ± 36.2 | |
| | | A/A | 78.9 ± 28.2 | 49.1 ± 26.3 | | 60.1 ± 26.7 | 100.1 ± 31.7 | |
| | rs7202116 | A/A | 79.7 ± 26.9 | 49.1 ± 23.8 | −/− | 60.1 ± 24.8 | 101.2 ± 38.3 | −/− |
| | | A/G | 80.3 ± 31.2 | 49.2 ± 24.1 | | 59.2 ± 22.1 | 99.6 ± 35.7 | |
| | | G/G | 78.2 ± 28.7 | 53.2 ± 27.2 | | 59.3 ± 26.2 | 100.2 ± 36.4 | |
| | rs9930506 | A/A | 78.5 ± 28.2 | 49.8 ± 23.8 | −/− | 61.2 ± 21.5 | 100.1 ± 39.7 | −/− |
| | | A/G | 81.2 ± 30.2 | 52.5 ± 26.5 | | 59.9 ± 22.3 | 99.2 ± 39.2 | |
| | | G/G | 82.1 ± 29.0 | 50.9 ± 24.1 | | 59.5 ± 25.6 | 98.9 ± 37.1 | |
| KCNJ11 | rs5219 | Glu/Glu | 80.1 ± 33.5 | 44.9 ± 19.2 | 0.020/0.044 | 46.2 ± 20.8 | 99.6 ± 37.5 | −/0.020 |
| | | Glu/Lys | 88.8 ± 32.2 | 53.2 ± 21.4 | | 43.7 ± 22.9 | 84.7 ± 38.2 | |
| | | Lys/Lys | 89.4 ± 31.2 | 54.2 ± 23.2 | | 43.7 ± 22.9 | 81.2 ± 39.9 | |
| SLC30A8 | rs13266634 | C/C | 78.4 ± 30.7 | 43.2 ± 17.7 | 0.030/0.018 | 48.3 ± 23.3 | 92.9 ± 41.1 | −/− |
| | | C/T | 88.9 ± 31.2 | 49.2 ± 22.7 | | 52.2 ± 26.7 | 96.2 ± 42.3 | |
| | | T/T | 89.8 ± 30.9 | 53.6 ± 19.1 | | 51.7 ± 22.5 | 93.6 ± 43.5 | |
| CDKN2B | rs10811661 | T/T C/T | 85.9 ± 31.4 | 49.4 ± 17.6 | 0.035/− | 47.9 ± 21.2 | 106.1 ± 34.7 | 0.021/0.042 |
| | | C/C | 82.4 ± 30.3 | 48.3 ± 16.5 | | 44.2 ± 20.1 | 95.2 ± 33.2 | |
| | | | 71.2 ± 34.5 | 48.7 ± 15.8 | | 32.1 ± 18.5 | 90.8 ± 29.9 | |
| CDKAL1 | rs7756992 | A/A | 82.4 ± 30.5 | 50.6 ± 20.1 | 0.033/0.045 | 60.8 ± 14.5 | 105.8 ± 38.8 | 0.023/0.041 |
| | | A/G | 79.9 ± 31.4 | 49.1 ± 19.4 | | 56.5 ± 21.0 | 99.9 ± 44.1 | |
| | | G/G | 71.8 ± 29.1 | 46.1 ± 21.1 | | 50.5 ± 21.9 | 96.6 ± 36.2 | |
| | rs9465871 | C/C | 85.1 ± 30.5 | 49.3 ± 24.1 | 0.025/0.035 | 53.0 ± 20.5 | 104.2 ± 48.2 | 0.021/0.041 |
| | | C/T | 80.5 ± 33.3 | 46.4 ± 22.9 | | 49.5 ± 23.9 | 97.0 ± 40.1 | |
| | | T/T | 71.8 ± 29.1 | 40.2 ± 19.2 | | 42.7 ± 18.9 | 96.0 ± 35.6 | |
| | rs7754840 | C/C | 80.1 ± 25.7 | 50.6 ± 22.6 | −/− | 60.4 ± 18.3 | 101.4 ± 39.4 | −/− |
| | | C/G | 79.9 ± 32.9 | 49.1 ± 22.7 | | 59.3 ± 20.4 | 99.3 ± 42.7 | |
| | | G/G | 79.7 ± 26.1 | 51.1 ± 25.5 | | 58.7 ± 24.7 | 101.8 ± 33.9 | |
| | rs10946398 | A/A | 85.7 ± 32.8 | 48.2 ± 17.7 | 0.032/0.047 | 60.2 ± 19.9 | 101.4 ± 39.4 | −/− |
| | | A/C | 83.2 ± 35.6 | 46.5 ± 20.2 | | 60.4 ± 21.3 | 99.3 ± 42.7 | |
| | | C/C | 72.4 ± 32.9 | 40.4 ± 18.5 | | 59.5 ± 24.2 | 101.8 ± 33.9 | |

of the *KCNJ11* gene showed statistically significant differences between the DM2+ and DM2− groups in the Russian population. The presence of the *Lys/Lys* genotype increased the risk of T2DM development (*OR* = 1.55), whereas that of the *Glu/Glu* genotype reduced development (*OR* = 0.65).

The protein of the *SLC30A8* gene plays a direct role in the maturation and secretion of insulin granules (*Dunn, 2005*). Previous work demonstrated that changes in this gene are associated with T2DM development in several populations (*Horikawa et al., 2008*; *Ng et al., 2008*; *Scott et al., 2007*). The SNP *rs13266634*, located in exon 8, has the most distinct association with diabetes. This SNP results in the replacement of arginine (R) by tryptophan (W) (*OR* = 1.12 in Caucasians) at position 325 of the protein sequence. The carriership of the 'predisposing-to-disease' allele *R325* is associated with a reduction in insulin secretion (also as a response to glucose stimulation (*Boesgaard et al., 2008*)) and impairment of the

transformation of proinsulin into insulin (*Kirchhoff et al., 2008*). Our study demonstrated an association between the SNP *rs13266634* of the *SLC30A8* gene with T2DM, with the *T/T* genotype as the predisposing genotype ($OR = 1.86$).

Previous studies have shown that the *CDKN2B/2A* gene plays a dual role in the deterioration of insulin secretion. The protein produce of this gene plays an indirect role in the regulation of *KCNJ11* gene expression by regulating *E2F1* gene expression, which in turn regulates *KCNJ11* gene expression (*Fajas, Blanchet & Annicotte, 2010*). It also participates in the regulation of β-cell proliferation (*Ferru et al., 2006*). Studies on the Chinese (*Kong et al., 2016*), African–American (*Lewis et al., 2008*), Japanese (*Omori et al., 2008*) and several European populations (*Grarup et al., 2007*; *Cauchi et al., 2008*; *Van Hoek et al., 2008*) have confirmed that polymorphisms at the *CDKN2A/2B* locus are associated with T2DM development. The *rs10811661* marker has the strongest association with diabetes in European populations ($OR = 1.19$) (*Cauchi et al., 2008*). We found that this polymorphic marker also had a strong association with T2DM in the Russian population ($OR = 2.10$).

Several polymorphisms (*rs7756992*, *rs7754840* and *rs10946398*) in the *CDKAL1* gene have exhibited association with T2DM (OR up to 1.15 in populations with European ethnicity) (*Dehwah, Wang & Huang, 2010*). Insulin secretion is reduced in response to glucose in carriers of the risk alleles *rs7756992* and *rs10946398* of the *CDKAL1* gene (*Pascoe et al., 2007*). To date, several SNPs have been identified in the *CDKAL1* gene that exhibit an association with low insulin secretion in individuals with and without T2DM, depending upon the population (*Wen et al., 2010*; *Hu et al., 2009*; *Tabara et al., 2009*; *Rong et al., 2009*). Three (*rs9465871*, *rs7756992* and *rs10946398*) out of the four examined polymorphic markers exhibited association with T2DM development in our population.

Insulin resistance is a major factor for T2DM development. An increase in body mass index (BMI) and fat mass contributes to the development and aggravation of immune resistance (*Kloting et al., 2008*; *Gerken et al., 2007*). Recent population studies have demonstrated that people who are homozygous for allele *A* of the *FTO* gene variant *rs9939609* have a higher BMI, weigh 3 kg more on average and are twice as likely to become obese compared to individuals homozygous for the protective allele *T/T* genotype (*De Luis et al., 2016*; *Livingstone et al., 2016*; *Munoz-Yanez et al., 2016*; *Moraes et al., 2016*; *Chen et al., 2017*). The presence of the protective *T* allele leads to increased lipolytic activity of adipocytes, thus reducing fat mass (*Wahlen, Sjolin & Hoffstedt, 2008*). Examinations of many patient populations have shown certain correlations between increased BMI, obesity and the presence of several SNPs, most notably *rs9939609* in intron 1 of the *FTO* gene ($OR = 1.42$ in individuals with European ethnicity) (*Hinney et al., 2007*).

We studied the effect of the polymorphic markers *rs8050136*, *rs7202116* and *rs9930506* (tag-SNP, characterising the linkage disequilibrium block in the promoter region) of the *FTO* gene on T2DM development. The analysis showed no statistically significant differences in the distribution of these polymorphic markers between the DM2+ and DM2− groups.

Based on these results, it can be concluded that the genes, *KCNJ11*, *SLC30A8*, *CDKN2B* and *CDKAL1*, affect the level of insulin synthesis and secretion in the β-cells of the pancreas

and play a significant role in T2DM development in the examined Russian population. The *FTO* gene associated with T2DM development in other populations is not associated with the disease in the Russian population. The results do not contradict previous research data, but the different OR values indicate that the contribution of different loci to T2DM development varies among different populations. It should be noted that these data are preliminary and require future confirmation using similar samples in independent studies.

The obtained data (OR and allele frequencies for polymorphic markers) will allow the quantitative assessment of the genetic risk of T2DM development in the Russian population. Understanding the genetic basis of disease development allows for better identification of the etiological mutations in the genes that determine susceptibility to T2DM. Understanding the mechanism underlying T2DM development should allow for development of new medications to protect against the development of this disease in genetically susceptible individuals.

We did not use a Bonferroni correction for multiple comparisons, which is a limitation of this study. However, we believe that adequate sample sizes and statistical significance of the comparisons will ensure the high reproducibility of the obtained results in future studies.

### Funding
The research was supported by the Russian Science Foundation (No. 14-25-00181). The funders had no role in study design, data collection and analysis, decision to publish, or preparation of the manuscript.

### Grant Disclosures
The following grant information was disclosed by the authors:
Russian Science Foundation: 14-25-00181.

### Competing Interests
The authors declare there are no competing interests.

### Author Contributions
- Aleksey G. Nikitin conceived and designed the experiments, analyzed the data, wrote the paper, prepared figures and/or tables, reviewed drafts of the paper.
- Viktor Y. Potapov conceived and designed the experiments, analyzed the data, contributed reagents/materials/analysis tools, wrote the paper, prepared figures and/or tables, reviewed drafts of the paper.
- Olga I. Brovkina and Dmitry S. Khodyrev performed the experiments, analyzed the data, contributed reagents/materials/analysis tools, reviewed drafts of the paper.
- Ekaterina O. Koksharova performed the experiments, contributed reagents/materials/-analysis tools, prepared figures and/or tables, reviewed drafts of the paper.
- Yury I. Philippov reviewed drafts of the paper.
- Marina S. Michurova and Viktor Y. Kalashnikov performed the experiments.

- Minara S. Shamkhalova and Olga K. Vikulova conceived and designed the experiments, performed the experiments.
- Svetlana A. Smetanina analyzed the data, prepared figures and/or tables, reviewed drafts of the paper.
- Lyudmila A. Suplotova, Olga M. Smirnova, Valery V. Nosikov and Alexander V. Averyanov conceived and designed the experiments.
- Irina V. Kononenko performed the experiments, wrote the paper, prepared figures and/or tables, reviewed drafts of the paper.
- Alexander Y. Mayorov performed the experiments, reviewed drafts of the paper.
- Marina V. Shestakova conceived and designed the experiments, wrote the paper, reviewed drafts of the paper.

## Human Ethics

The following information was supplied relating to ethical approvals (i.e., approving body and any reference numbers):

Local Committee for Ethics of Endocrinology Research Centre (Moscow, Russian Federation) granted ethical approval to carry out the study (Ethical Application Ref: protocol No.14AB on 27-nov-2014).

## Data Availability

Nikitin, Alex; Potapov, Viktor; Brovkina, Olga; Koksharova, Ekaterina; Khodyrev, Dmitry; Philippov, Yury; Michurova, Marina; Shamkhalova, Minara; Vikulova, Olga; Smetanina, Svetlana; Suplotova, Lyudmila; Kononenko, Irina; Kalashnikov, Viktor; Smirnova, Olga; Mayorov, Aleksander; Nosikov, Valery; Averyanov, Alexander; Shestakova, Marina (2016), "Polymorphic markers of genes *FTO*, *KCNJ11*, *CDKAL1*, *SLC30A8*, and *CDKN2B* in Russian population of Type 2 Diabetes", Mendeley Data, v2 http://dx.doi.org/10.17632/fys583ghzm.2.

## Supplemental Information

Supplemental information for this article can be found online at http://dx.doi.org/10.7717/peerj.3414#supplemental-information.

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
