# Peer review of "Association of polymorphic markers of genes *FTO, KCNJ11, CDKAL1, SLC30A8,* and *CDKN2B* with type 2 diabetes mellitus in the Russian population"

_PeerJ, doi:10.7717/peerj.3414_

## Round 0.1 · original submission · Major Revisions

Please carefully address the reviewer's comments.

Reviewer 1 ·

Basic reporting

- The Introduction is too long and a little vague. I would suggest the authors to shorten this section and to put their study in the current context briefly.

- The discussion section is missing. Authors should discuss their results, compare with previous findings from the literature, and state the limitations/strengths of their study.

Experimental design

- The description of the Genes/SNPs selection is missing in the Materials and Methods section. The authors should explain why they tested association with T2D in only these 5 specific genes and how they selected the 10 SNPs.

- I thank you the authors for providing the raw data, however the Materials and Methods section need details regarding the genotyping and quality control. What were the genotype call rates for each SNPs ? Any failed samples ? Were any samples used as control duplicates ?

Validity of the findings

- No multiple testing correction has been applied (e.g. Bonferonni correction) for the number of SNPs and phenotypes tested.

- Results are not convincing enough to draw conclusions. A discussion would have been useful to state the limitations of this study (limited sample size, lack of replication in an independent sample) and to moderate the conclusions.

·

Basic reporting

English needs revision. For example sentences like " [...] we performed an analysis of the distribution of frequencies polymorphic markers of these genes" or "The key causes of T2DM pathogenesis include insulin resistance" need to be rephrased.

Concepts expressed in paragraphs at lines 51-66 and 67-75 need to be supported by references. Moreover, references 2-3, supporting the statement that about 30 genes have been associated with T2D, are too old (2009). The current list of genes is larger and authors should cite a more recent reference.

Experimental design

Authors should clearly justify their choice of SNPs to test for potential association with T2D in the Russian population. Why some SNPs and not some others? For example: is the overall state of the art about the chosen SNPs less consistent compared to the other SNPs? Or perhaps the chosen SNPs have never been tested in the Russian population whereas the others have, and so on...Despite a long introduction with detailed descriptions of the genes containing the chosen SNPs, and of previous data existing on these SNPs, the reader is not allowed to understand why the authors decided to assess, in a Russian population, the associations they assessed.

It is not clear how healthy controls were recruited. Authors speak of "health resort patients" (line 182) without further details.

Validity of the findings

Authors should avoid to state that "In the Russian population, genes affecting the level of synthesis and secretion of insulin in the β-cells of the pancreas play a central role in the development of T2DM." and they should rather state that "[...] SNPs in genes [...] are associated with T2DM.".

Additional comments

The introduction is really too long. All the sentences on specific gene functions are not required and, as above mentioned, the authors should rather better clarify why they performed the proposed study.
All the details on previous data existing on the tested SNPs and T2DM risk are interesting but they should be moved to the discussion/conclusion, contributing to the interpretation of the study's results.

Reviewer 3 ·

Basic reporting

The authors present results of a replication study in subjects of Russian origin for 10 SNPs associated to Type 2 diabetes. Unfortunately, the literature cited in their paper, although relevant and quite numerous, is completely outdated. Not a single reference to papers published after 2010! A recent review on SNPs and genes associated to Type 2 diabetes and glycemic traits enumerates around 90 loci for T2D only! (see Mohlke KL, Boehnke M. Recent advances in understanding the genetic architecture of type 2 diabetes. Hum Mol Genet. 2015;24(R1):R85–92).

Experimental design

Results are not novel, at best the study replicates some loci in a Russian sample for SNPs known to be associated in many other population samples of different origin. Statistical tests are not the most currently employed in genetic epidemiology, even if they could be considered adequate. Also, one thing that stroke me is the fact that there are no missing genotypes for the 10 SNPs assayed on 1305 subjects.

Validity of the findings

Replication is always welcomed in genetics of metabolic disorders, but some of the reported analyses are absolutely useless. In Table 4, the authors describe results of association for some traits (insulin levels and HOMA-B) which appear to be genetically linked to the selected 10 SNPs. The problem is that these traits are known to be different between diabetic and non-diabetic people! Their analysis does not test association between these SNPs and the traits in question, but only confirms what is known to everyone in endocrinology.

Additional comments

Literature update is absolutely required before this paper has a single chance of being published in any serious journal.

---

## Round 0.2 · Major Revisions

Please carefully address the remaining reviewer's comments.

Reviewer 1 ·

Basic reporting

The writing of the manuscript is much improved.
The Introduction has been shorten, and a discussion has been added, however the limitations/strengths of this study are still not discussed.

Experimental design

- A description of the Gene/SNP selection has been added in the Materials and Methods section, however it is somewhat vague.

- Information regarding the genotyping and quality control is also too vague "Samples that failed to meet this quality value were reanalyzed". How many samples failed ? Did those passed QC after reanalyzed ?

Validity of the findings

A multiple testing correction should be applied (or at least, the authors should discuss this point as a limitation of the study).

Additional comments

While the manuscript has been improved, the authors partially addressed my concerns.
Further, a rebuttal letter providing detailed responses addressing the individual comments of the reviewers would have been useful.

Reviewer 3 ·

Basic reporting

The authors made a real effort in updating the scientific literature relevant to their study.
English language needs to be revised as it does not conform to professional standards. Bad syntax and typos remain.
In Table 1, most values of means and standard deviations have changed from the 1st revision. Please explain why.
Table 2 should be moved to Supplementary Tables.
“Prevalence” must be substituted for “incidence” in many places in the main text.
Be consistent in using either “beta cells” or “β-cells”.
In Table 4, what means “ND”?

Experimental design

No comment

Validity of the findings

In the Discussion, based on the results of your study, you cannot claim that “a polymorphic marker contributes to the development of T2DM”. You can only say that your analysis revealed an association between the marker and T2DM. The former claim can only be made with longitudinal data.

---

## Round 0.3 · accepted · Accept

The second revision of the manuscript is suitable for publication.